# Peri- and Interprosthetic Femoral Fractures—Current Concepts and New Developments for Internal Fixation

**DOI:** 10.3390/jcm11051371

**Published:** 2022-03-02

**Authors:** Clemens Kösters, Daniel den Toom, Sebastian Metzlaff, Kiriakos Daniilidis, Linda Barz, Steffen Roßlenbroich

**Affiliations:** 1Klinik für Orthopädie, Unfall- und Handchirurgie, Maria-Josef-Hospital Greven, 48268 Greven, Germany; daniel.dentoom@mjh-greven.de (D.d.T.); linda.barz@mjh-greven.de (L.B.); 2Klinik für Orthopädie und Unfallchirurgie, St. Joseph Krankenhaus Berlin, 12101 Berlin, Germany; sebastian.metzlaff@sjk.de; 3Orthopädie Traumatologie Centrum Regensburg, 93049 Regensburg, Germany; daniilidis@otc-regensburg.de; 4Klinik für Unfall-, Hand- und Wiederherstellungschirurgie, Universitätsklinikum Münster, 48149 Münster, Germany; steffen.rosslenbroich@ukmuenster.de

**Keywords:** periprosthetic fractures, current concepts, elderly patients, plate osteosynthesis

## Abstract

Treatment of peri- and interprosthetic fractures represents a challenge in orthopedic trauma surgery. Multiple factors such as osteoporosis, polymedication and comorbidities impede therapy and the rehabilitation of this difficult fracture entity. This article summarizes current concepts and highlights new developments for the internal fixation of periprosthetic fractures. Since the elderly are unable to follow partial weight bearing, stable solutions are required. Therefore, a high primary stability is necessary. Numerous options, such as new angular stable plate systems with additional options for variable angle screw positioning, already exist and are in the process of being further improved. Lately, individually produced custom-made implants are offering interesting alternatives to treat periprosthetic fractures.

## 1. Introduction

Joint replacement is a successful and standardized method to treat osteoarthritis and joint fractures that cannot be reconstructed. In correlation with the increasing number of total joint replacements, periprosthetic fractures have also become more relevant.

Although patients demand to maintain high levels of physical activity [1], the severity of the fracture, combined with the usually preexisting multimorbidity, worsens the outcome [2]. The main goal of treatment is to achieve high postoperative stability allowing early mobilization and full weight bearing.

The lifetime prevalence for patients in Germany to suffer from osteoarthritis is about 25% for women and about 16% for men. These data were compiled in a representative survey of the adult population of Germany [3].

In 2011, Della Rocca et al. described an increasing incidence of periprosthetic fractures due to a higher life expectancy and higher activity levels in advanced age [4]. Today, periprosthetic fractures are not just a problem in the trauma surgery of the elderly but a topic for the “best agers” [5].

Most commonly, periprosthetic fractures affect the femur [6]. They can further be divided into intraoperative iatrogenic fractures and traumatic fractures [7].

The rate of intraoperative periprosthetic femur fractures during (mostly) primary cementless femoral shaft implantation is about 0.1–1.0%. However, in revision surgeries these numbers increase dramatically to up to 6% [8]. In traumatically caused periprosthetic fractures, aseptic loosening of the implant and non-press fit implanted cement free stems are the main risk factors [9].

Regarding traumatic periprosthetic fractures, the data differ for total knee arthroplasties (TKA) (0.3–5.0%) [8,10] and total hip arthroplasties (THA) (0.1–6.0%) [11,12]. The highest risk for periprosthetic fractures is a press-fit implantation of THR (up to 5%) and revisions of TKA and THA (up to 20%) [4,11].

Comorbidities, such as chronic peptic gastrointestinal lesions or chronic obstructive lung diseases—especially after corticosteroids—were indicated to increase the number of periprosthetic fractures [12]. Additional recently published studies show neurological diseases (e.g., parkinsonism or a stroke) further increase the incidence of periprosthetic fractures [13]. Other new studies have pointed out a higher risk for hip fractures to occur in the elderly in correlation with the number of drugs [14].

## 2. Characteristics and Classification

The Vancouver classification for periprosthetic hip fractures and the Lewis and Rorabeck classification for periprosthetic knee fractures remain the most accessible classification systems [15].

The Vancouver classification proposed by Duncan and Masri [16] takes the surgically most relevant factors, such as the quality of the surrounding bone stock and the condition of the prosthesis, into account. Vancouver type A describes a fracture located in the proximal trochanter region. An additional letter is added depending on whether the greater (G) or lesser trochanter (L) is affected.

Vancouver type C usually covers all periprosthetic fractures that are below the stem of the prosthesis. Therefore, both Vancouver type A and C, result in a stable prosthesis.

With an incidence of 88% of all periprosthetic hip fractures [17], Vancouver type B includes all periprosthetic fractures around the stem. Being divided into three different sub-categories, type B fractures are usually tied to a certain surgical treatment:

B1: stable prosthesis, operative stability check

→if stable: locking plate osteosynthesis→if unstable: treated as B2

B2: unstable prosthesis with good bone stock

→long-stem or modular revision prosthesis

B3: unstable prosthesis with poor bone stock

→modular revision or tumor prosthesis

Further systems exist to categorize periprosthetic fractures. The recent system to classify all periprosthetic fractures by Duncan and Haddad called the ‘Unified Classification System’ (UCS) is geared to the principles of the AO classification and links them to the Vancouver cataloging criteria. It is suitable for all anatomic localizations and every bone [18]. The classification code consists of three parts: the first is a Roman numeral corresponding to the joint localization from top to bottom (shoulder = 1, elbow = 2, wrist = 3, hip = 4, knee = 5, ankle = 6); second is the Arabic numeral of the affected bone’s AO classification; the last part is a specification of the fracture, following the Vancouver classification. Depending on the type (Table 1), Duncan and Haddad suggest a certain treatment algorithm.

## 3. Preoperative Planning

The strategy of treatment of peri- and interprosthetic fractures heavily depends on the status and stability of the implants. Although intraoperative assessment is consistently performed, the careful preoperative evaluation of the symptoms for loosening prior to the fracture is mandatory for an adequate treatment.

The following clinical signs may indicate loosening of the prosthesis: pain during rest or movement, swelling or articular effusion, limited mobility or unsteady gait, length reduction or axial deviation and any clinical signs of infection.

Generally, reviewing recent radiographs is advisable to further identify signs of loosening such as osteolysis, axial deviation or bony erosion around the prosthesis.

Besides conventional X-rays, CT scans should be taken of each fracture to allow a detailed fracture analysis and classification. Digital surgical planning software is recommended to determine the intraoperatively used implants and sizes as exactly as possible prior to surgery. In case of clinical signs of infections, a joint puncture should be performed.

## 4. Treatment of Periprosthetic Fractures

Although conservative treatment should be reserved for special cases, the treatment of choice depends on the stability of the prosthesis. An implant that appears to be unstable, either due to loosening before the trauma or because of the fracture, should be changed depending on the remaining bone stock.

In a stable prosthesis, however, osteosynthesis is the treatment of choice.

Currently the surgeon has the choice of several plate systems which were developed especially for the treatment of periprosthetic fractures. To allow for better blood supply, angle-stable plates have proven superior due to a smaller contact area. Chatiagorou et al. showed a significantly lower revision rate in these plates compared to regular ones [19].

So far, the most frequently applied plates are on the one hand the ‘Variable angle locking compression plate’ (VA-LCP^®^ condyle plate) by Depuy-Synthes, and on the other hand, the ‘Noncontact bridging-periprosthetic plate’ (NCB-PP^®^) designed by Zimmer-Biomet. The NCB-PP^®^ offers opportunities adjusted to the anatomical variability of the fracture region of the femur.

Recently two new periprosthetic plate systems were launched. The 3.5/4.5 Variable Angle LCP^®^ Periprosthetic Proximal Femur Plating System by Depuy-Synthes provides two main proximal plate options and three additional attachment plates with dedicated plate features (Figure 1) [20].

The EVOS^®^ Large & Periprosthetic Plating System by Smith & Nephew offers anatomically preformed broad plates with multiple variable angle screw options to operate proximal and distal periprosthetic femur fractures (Figure 2).

All the systems can be inserted minimally invasively and if necessary, can be combined with composition plates or cable cerclages.

The major operative challenge is the bicortical fixation of the screws around the enclosed stem of the prosthesis.

Hence, a special concept improving the bicortical screw fixation around intramedullary devices called the LOQTEQ^®^ VA Periprosthetic Plate or ‘periprosthetic hinge plate’ was established by the company, aap Implantate AG. Multiple insertable hinges create the possibility to place the screws bicortically and alongside the enclosed stem. The square of the two polyaxial locking screws of the hinge constitutes 15 degrees to each direction. Since a central screw can be attached separately, further angle stabilization of the hinges can be achieved. The hinge is not attached to a plate hole, which could be staffed additionally with monocortical screws or cable cerclages in the diaphysis (Figure 3).

A biomechanical study investigated the characteristics of the LOQTEQ^®^ VA Periprosthetic Plate in comparison with the standard locking compression plate with locking attachment plate, for the treatment of periprosthetic fractures in a Vancouver B1 fracture model.

The LOQTEQ^®^ VA Periprosthetic Plate showed superior biomechanical results (axial stiffness and cycles to failure) compared to the standard locking compression plate in combination with the locking attachment plate [21].

### 4.1. Double Plating

In addition to the specifically designed plates, the possibility of a double plate osteosynthesis should be taken into consideration. The internal fixation with two plates increases the primary stability and gives the patient the opportunity for earlier full weight bearing [22].

Biomechanical studies proved the advantages of double plating over one plate, even in combination with strut-graft fixation [23]. Shin et al. demonstrated faster radiographic bone union in periprosthetic fractures treated with double plates compared to a single plate [24].

In revision cases and interprosthetic fractures, double plating is an alternative treatment option to avoid high-risk operations such as a total femur replacement. Figure 4 shows a successfully treated hypertrophic non-union after several interprosthetic re-fractures, using double plating in combination with autologous bone-grafting and growth factor application.

### 4.2. Additional Cerclage

The additional application of cerclages still remains a debate in the recent literature. The braided cable cerclage wins over the common wire cerclage, with its easy handling, but is much more expensive. Biomechanical studies illustrate the ideal handling of cerclage [25]. If applied as a double wrapped cerclage, the stability increases even further [26].

### 4.3. Cement Augmentation

In the last few years, cement augmentation, especially in osteoporotic bones, became another surgical option to increase the primary stability and to reduce secondary complications, such as loss of reposition or ‘cut out’. Clinical follow up studies need to be put into practice. Biomedical studies already illustrated the advantages in osteoporotic bone [27,28].

### 4.4. Interposition Sleeves, Docking Tools and Custom-Made Implants

Although rare, interprosthetic fractures require cautious preoperative planning. Depending on the quality of the bone stock and the implant stability, either osteosynthesis via locked plates, the use of special devices such as docking tools (“Osteobridge^®^”, Merete Medical, Berlin, Germany) or custom-made interposition sleeves (AQ-Implants GmbH, Ahrensburg, Germany; “RescueSleeve^®^”, Waldemar Link GmbH, Hamburg, Germany) should be considered to avoid revision total hip replacement (RTHR) or a total femur implantation.

The “Osteobridge^®^” is a modular system which spans the fracture according to the distal and proximal end, respectively, over the implant stems of the endoprosthesis [29]. Interposition devices, such as the “RescueSleeve^®^”, are used to replace the fractured diaphyseal bone and to couple both ends of the hip and knee stem using two screws connecting the sleeves (Figure 5) [30], while custom-made sleeves could be used to couple a stable implant to a tumor endoprosthesis (AQ-Implants GmbH, Ahrensburg, Germany) (Figure 6).

## 5. Conclusions

Peri- and interprosthetic fractures remain a big challenge and a high skill level in both osteosynthesis and revision arthroplasty is needed to treat these injuries. However, full weight bearing should be the main therapeutic objective in every patient. A broad variety of specially developed implants and plate systems provide many options to reach a high primary stability of internal fixation. An interdisciplinary approach is needed to optimize the therapy addressing comorbidities and to secure early rehabilitation under full weight bearing.

## Figures and Tables

**Figure 1 jcm-11-01371-f001:**
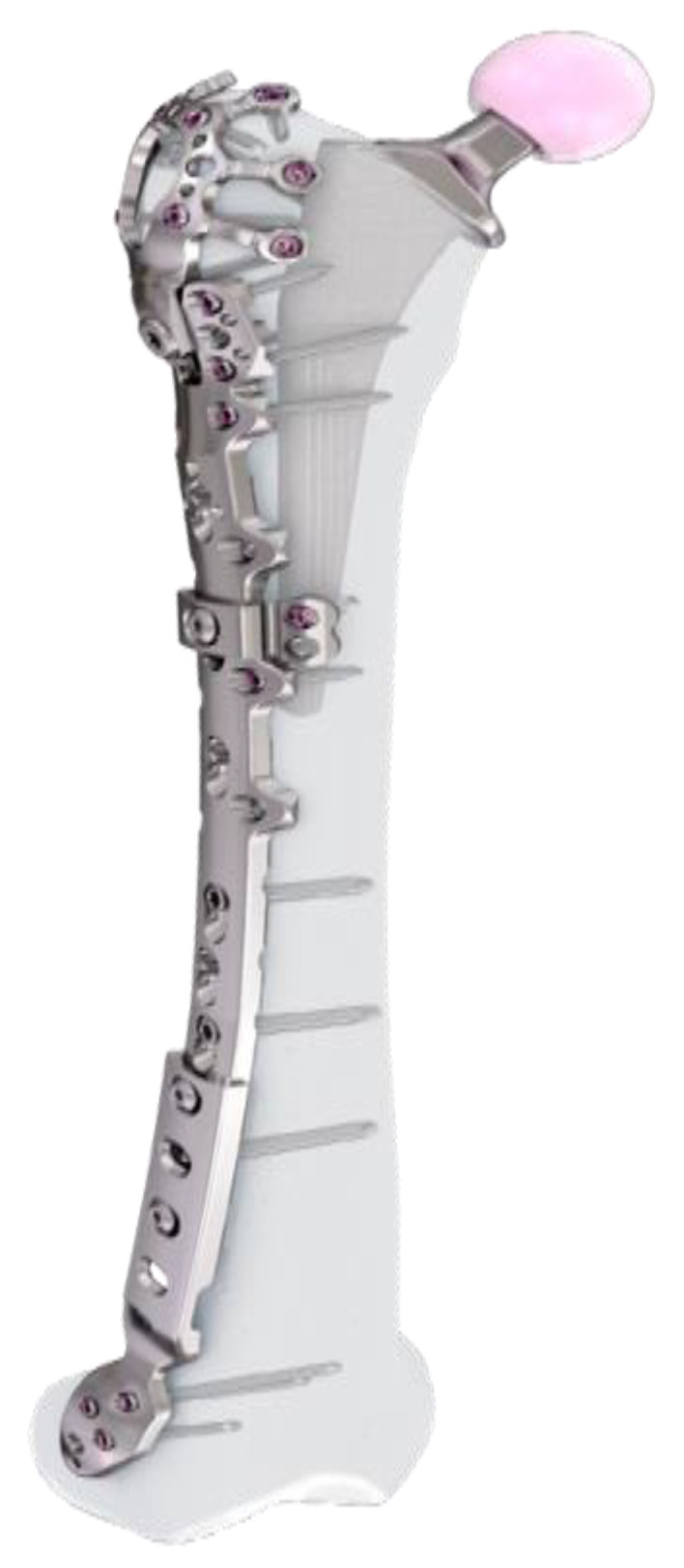
The 3.5/4.5 Variable Angle LCP^®^ Periprosthetic Proximal Femur Plating System by Depuy-Synthes from [20]. (Reprinted with permission from [20] 2021 Depuy-Synthes).

**Figure 2 jcm-11-01371-f002:**
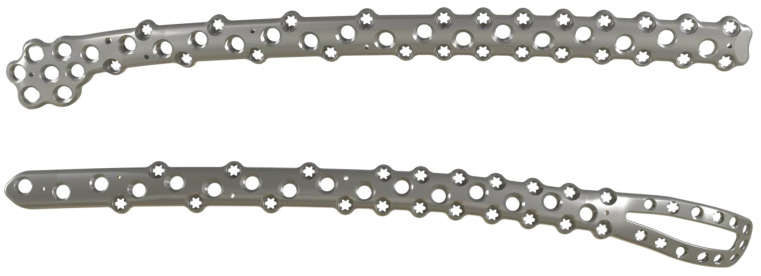
Distal and proximal plates of the EVOS Large & Periprosthetic Plating System by Smith & Nephew.

**Figure 3 jcm-11-01371-f003:**
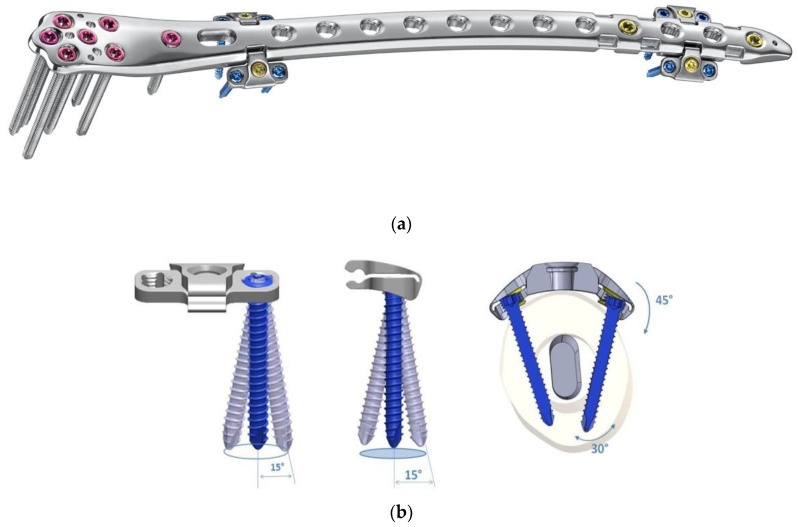
(**a**) LOQTEQ VA Periprosthetic Plate (with kind permission of aap Implantate AG, Berlin, Germany). (**b**) Insertable hinge with each of the two variable angle screw options (with kind permission of aap Implantate AG, Berlin, Germany). (**c**) Periprosthetic fracture around a cemented stable total hip revision arthroplasty. (**d**) LOQTEQ VA Periprosthetic Plate with mounted aiming device. (**e**) LOQTEQ VA Periprosthetic Plate with four inserted hinges. (**f**) Variable angle screw placement around the hip revision stem. (**g**) Postoperative X-rays. Four hinges with each of the two variable angle screw options were used to fix the plate around the hip stem.

**Figure 4 jcm-11-01371-f004:**
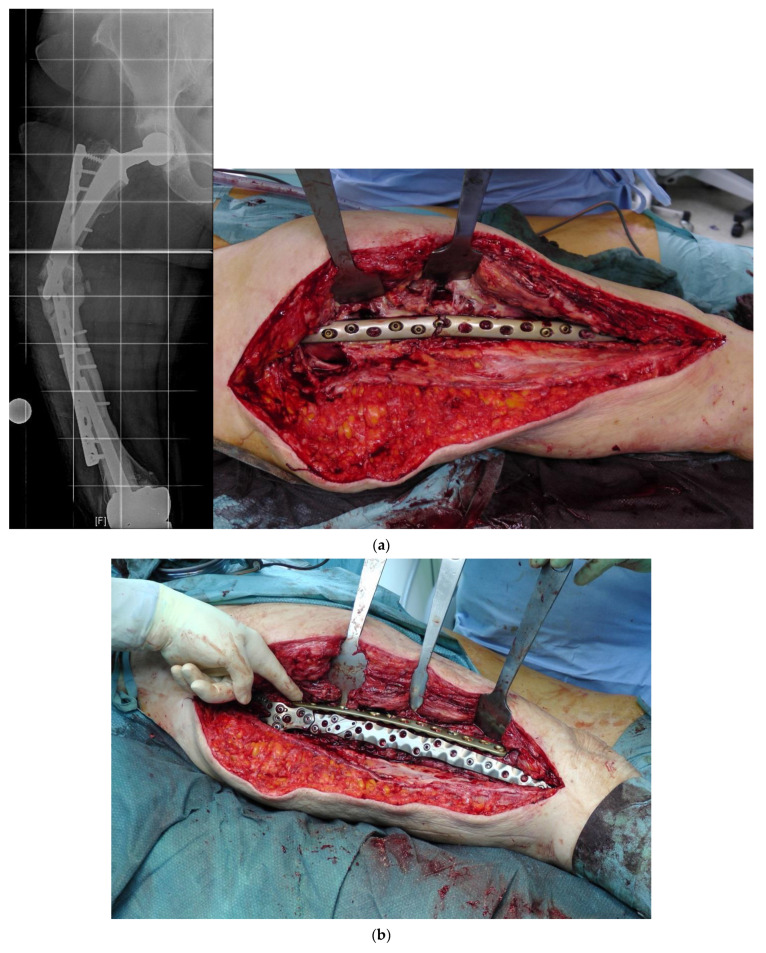
(**a**) Interprosthetic hypertrophic non-union. Intraoperative situs showing the failed plate fixation and the non-union. (**b**) Intraoperative situs showing the double plating. (**c**) Double plating combined with bone-grafting and growth factor application. (**d**) Postoperative follow-up X-rays after 6 weeks, 12, and 24 months, demonstrating complete healing and remodeling of the fracture.

**Figure 5 jcm-11-01371-f005:**
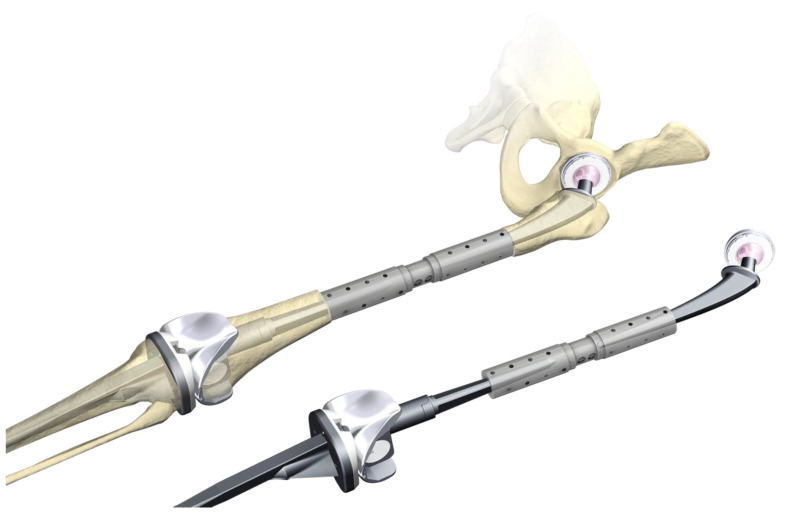
Custom-made interposition sleeve (Waldemar Link GmbH, Hamburg, Germany).

**Figure 6 jcm-11-01371-f006:**
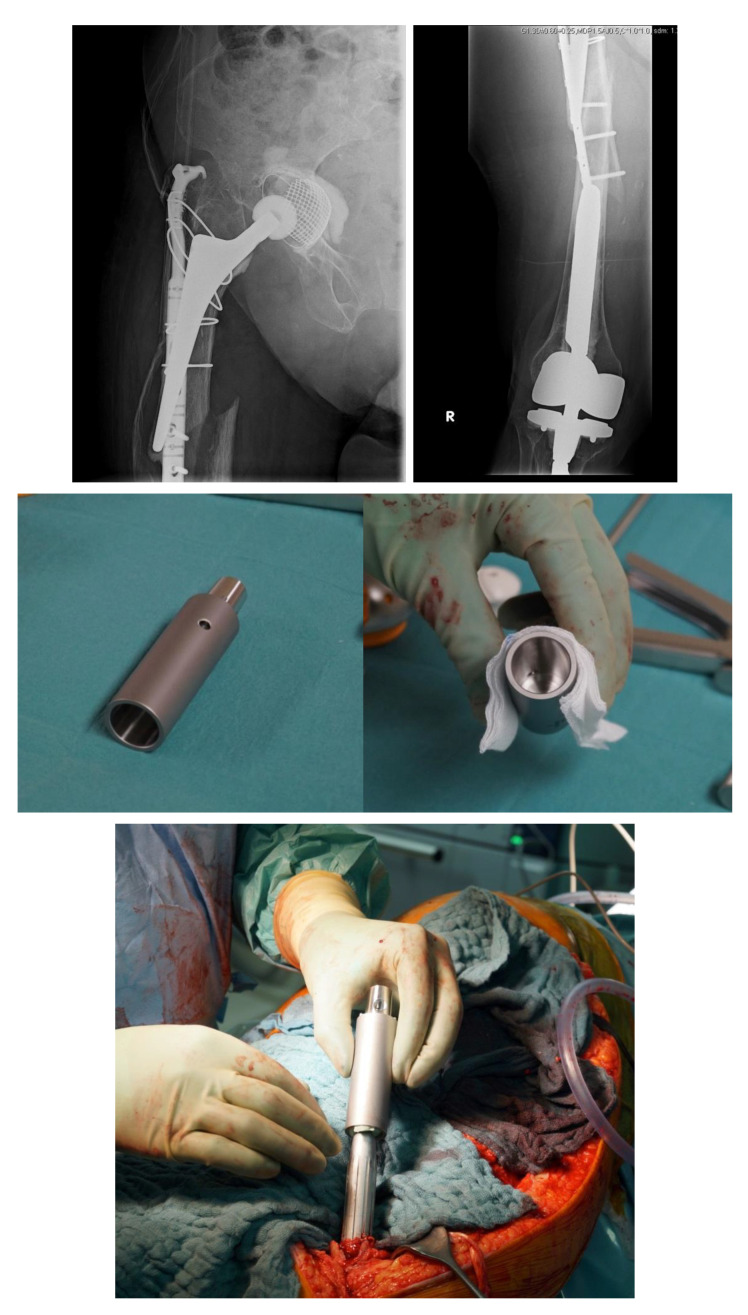
Interprosthetic fracture around a stable total knee revision arthroplasty and a loose total hip revision arthroplasty. Implantation of a custom-made docking sleeve coupled to a proximal femur replacement (AQ-Implants GmbH, Ahrensburg, Germany).

**Table 1 jcm-11-01371-t001:** Principles of UCS Classification as proposed by Duncan and Haddad—Third part of the classification code.

Type	Characteristics	Stability
A	Fracture of apophysis or tuberositas	+
B	Fracture of the region of the enclosed prosthesis I. normal bone quality II. normal bone quality III. Poor bone quality	+ − −
C	Fracture next to the prosthesis (proximal or distal)	+
D	Interprosthetic fracture between two enclosed prostheses	+/−
E	Fracture of two bones which build the base of a prosthesis	+/−
F	Fracture of joint surface which articulates with the prosthesis	+

## Data Availability

Data supporting reported results can be provided by the authors.

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
