# Peer review of "Peri- and Interprosthetic Femoral Fractures—Current Concepts and New Developments for Internal Fixation"

_jcm, 2022, doi:10.3390/jcm11051371_

Round 1

Reviewer 1 Report

A comprehensive review of PPF around femur. But there were still somewhere need to revision:

  1. Page 2, about the classification: PPF around knee, "The Lewis and Rorabeck classification is divided into type 1 for stable and not dislocated fractures, type 2 for dislocated but stable fractures and type 3 for loosened and dislocated or not dislocated. [16]".

         I suggest delete the sentence. Because all the discussion is focus on               PPF around hip stem, but not total knee arthroplasty.

  1. Figure A1, please edit the figure to cut off left lower corner “word”; the right side of figure is low resolution, please re-edit; and please indicate the source of the figure.
  2. Figure A2, same as figure 1: the right side of figure is low resolution, please re-edit; and please indicate the source of the figure.
  3. Figure A3 & A4, please number the order of each illustration, such as “a,b,c…”, and brief a description of the illustration, in order to make easier readability.
  4. Page 8, "A biomechanical study investigated the characteristics of this newly developed plate and hinge construct in comparison to the standard locking compression plate with locking attachment plate for the treatment of periprosthetic fractures in a Vancouver B1 fracture model. Additionally, the screw placement in the hinge was compared to the LAP regard-ing intraoperative handling.
    The newly developed LOQTEQ® VA Periprosthetic plate showed superior biome-chanical results (axial stiffness and cycles to failure) compared to the standard locking compression plate in combination with the locking attachment plate. Furthermore, the handling test showed a way easier implant fixation around the hip stem in periprosthetic fractures using the new variable angle hinge plate.[22]"

         Please condense this paragraph, to make more readability.

Author Response

Dear Reviewer,

Thank you for the opportunity to revise our manuscript.

As requested in the following you find our responses to your comments.

Kind regards,

The Authors

COMMENTS:

Reviewer #1: General comments
A comprehensive review of PPF around femur. But there were still somewhere need to revision:

Thank you.

Page 2, about the classification: PPF around knee, "The Lewis and Rorabeck classification is divided into type 1 for stable and not dislocated fractures, type 2 for dislocated but stable fractures and type 3 for loosened and dislocated or not dislocated. [16]".

I suggest delete the sentence. Because all the discussion is focus on PPF around hip stem, but not total knee arthroplasty.

We deleted this sentence!

Figure A1, please edit the figure to cut off left lower corner “word”; the right side of figure is low resolution, please re-edit; and please indicate the source of the figure.

We re-edited the figure and changed to an image with high resolution

Figure A2, same as figure 1: the right side of figure is low resolution, please re-edit; and please indicate the source of the figure.

We re-edited the figure and changed to an image with high resolution

Figure A3 & A4, please number the order of each illustration, such as “a,b,c…”, and brief a description of the illustration, in order to make easier readability.

We numbered each illustration from A3a to A3g, A4a to A4d and added a brief description of each illustration

Page 8, "A biomechanical study investigated the characteristics of this newly developed plate and hinge construct in comparison to the standard locking compression plate with locking attachment plate for the treatment of periprosthetic fractures in a Vancouver B1 fracture model. Additionally, the screw placement in the hinge was compared to the LAP regard-ing intraoperative handling.
The newly developed LOQTEQ® VA Periprosthetic plate showed superior biome-chanical results (axial stiffness and cycles to failure) compared to the standard locking compression plate in combination with the locking attachment plate. Furthermore, the handling test showed a way easier implant fixation around the hip stem in periprosthetic fractures using the new variable angle hinge plate.[22]"

         Please condense this paragraph, to make more readability.

We condensed this paragraph as follows:

“A biomechanical study investigated the characteristics of the LOQTEQ® VA Periprosthetic plate in comparison to the standard locking compression plate with locking attachment plate for the treatment of periprosthetic fractures in a Vancouver B1 fracture model.

The LOQTEQ® VA Periprosthetic plate showed superior biomechanical results (axial stiffness and cycles to failure) compared to the standard locking compression plate in combination with the locking attachment plate. [21]”

Reviewer 2 Report

a good update about a common clinical problem, however, the review lacks organization and smoothness of flow.

I have highlighted some areas for improvement.

line 31: All the more should not be used in the beginning of the sentence, remove or change.

line 33: it is better to put percentages instead of using quarter and 1/6

line 36,37: rephrase.

line 48: the highest

line 49: change to ''press-fit implantation of THR'' 

line 50: revisions of what THR or TKR or both?

line 94: why the section is numbered 3.1 when there is no 3.2, it should be 3 only.

94 to 103: Section 3 about preoperative planning is missing a lot of steps for instance:  Imaging studies, surgical planning, exclusion of infection etc. this requires improvement.

109 and 111: change therapy to treatment

section 4 and subsections are about operative treatment, subsections 4.1, 4.2, etc should be titled either according to the type of fracture treated or by the fixation technique used.

173: remove especially

189: Correct braded to braided.

In general, the review mentioned briefly revision surgeries and proximal or distal femoral replacements, it should have been mentioned in more detail.

Author Response

Reviewer #2: a good update about a common clinical problem, however, the review lacks organization and smoothness of flow.

I have highlighted some areas for improvement.

Thank you.

line 31: All the more should not be used in the beginning of the sentence, remove or change.

We removed “All the more”:

“The main goal of treatment is to achieve high postoperative stability allowing early mobilization and full weight bearing.”

line 33: it is better to put percentages instead of using quarter and 1/6

We changed to percentages as follows: “The lifetime prevalence for patients in Germany to suffer from osteoarthritis is about 25% for women and about 16% for men.”

line 36,37: rephrase.

We rephrased as follows:

“In 2011 Della Rocca et al. described an increasing incidence of periprosthetic fractures due to higher life expectancy and higher activity levels in advanced age.”

line 48: the highest

We changed to “The highest risk for periprosthetic fractures is a cementless press fit implantation (up to 5%) and revisions (up to 20%). [5,11]”

line 49: change to ''press-fit implantation of THR'' 

We changed to “The highest risk for periprosthetic fractures is a press-fit implantation of THR (up to 5%) and revisions (up to 20%). [5,11]”

line 50: revisions of what THR or TKR or both?

We added: “The highest risk for periprosthetic fractures is a press-fit implantation of THR (up to 5%) and revisions of TKA and THA (up to 20%). [5,11]”

line 94: why the section is numbered 3.1 when there is no 3.2, it should be 3 only.

We changed to “3. Preoperative planning”

94 to 103: Section 3 about preoperative planning is missing a lot of steps for instance:  Imaging studies, surgical planning, exclusion of infection etc. this requires improvement.

We added: “Besides conventional x-rays CT-scans should be taken of each fracture to allow a detailed fracture analysis and classification. Digital surgical planning software is recommended to determine the intraoperatively used implants and sizes as exact as possible prior to surgery. In case of clinical signs of infections a joint puncture should be performed.”

109 and 111: change therapy to treatment

We changed therapy to treatment!

section 4 and subsections are about operative treatment, subsections 4.1, 4.2, etc should be titled either according to the type of fracture treated or by the fixation technique used.

We changed to “4.3. Interposition sleeves, docking tools and custom-made implants”

173: remove especially

We removed “especially”

189: Correct braded to braided.

We corrected “braded to braided”

In general, the review mentioned briefly revision surgeries and proximal or distal femoral replacements, it should have been mentioned in more detail.

The aim of this review article was to summarize current concepts and new developments for internal fixation, therefore we just briefly mentioned implant revision surgeries and replacements. In our opinion a detailed description of new developments of revision arthroplasties/replacements would be to extensive for this review.

Round 2

Reviewer 2 Report

Thanks for the corrections